# Chromatin Accessibility Dynamics Reveal Conserved Transcriptional Regulatory Networks During Insect Metamorphosis in *Harmonia axyridis* and *Drosophila melanogaster*

**DOI:** 10.3390/biology14080912

**Published:** 2025-07-22

**Authors:** Jiejing Tang, Hang Zhou, Ziqi Cheng, Yang Mei, Yueqi Lu, Xi Chen

**Affiliations:** 1Institute of Biotechnology, Zhejiang University, Hangzhou 310058, China; tangjiejing@zju.edu.cn (J.T.); zhouhang716@zju.edu.cn (H.Z.); 2College of Plant Protection, Jilin Agricultural University, Changchun 130118, China; chengziqi1004@gmail.com (Z.C.); meiyang12@zju.edu.cn (Y.M.); 3Yangtze Delta Region Institute (Quzhou), University of Electronic Science and Technology of China, Quzhou 324000, China; yueqi_lu@csj.uestc.edu.cn; 4Department of Clinical Laboratory, The Quzhou Affiliated Hospital of Wenzhou Medical University, Quzhou People’s Hospital, Quzhou 324000, China

**Keywords:** insect metamorphosis, chromatin accessibility, ATAC-seq, transcriptional regulation, gene regulatory networks

## Abstract

Insects undergo dramatic transformations, known as metamorphosis, changing from larvae to adults. This process is tightly regulated by genes and hormones. However, scientists do not fully understand how these genes are controlled during metamorphosis. In our study, we investigated how the accessibility of DNA—which determines how easily genes can be switched on or off—changes throughout insect development. Using two insects (a ladybird beetle and a fruit fly) as models, we identified periods when DNA was especially open, allowing important genes to become active. We found a group of key genes common to both insects, suggesting they play fundamental roles in insect metamorphosis. Among these, we highlighted four major genes acting as central switches to control this process. Our results help explain how insects precisely manage their dramatic developmental changes. Understanding these mechanisms is important because it could help scientists develop new methods to manage insect populations, benefiting agriculture by improving pest control, and aiding in the conservation of beneficial insect species.

## 1. Introduction

Insects constitute one of the most abundant and diverse biological groups on Earth, with their evolutionary success largely attributed to the development of complete metamorphosis as a life history strategy used by approximately 80% of insect species [1]. This developmental process encompasses a series of complex morphological and physiological transformations, progressing from the egg to larva, pupa, and adult stages, each characterized by distinct morphological features and biological functions [2]. These dramatic developmental transitions are orchestrated through hormonal regulation and precise gene expression programs [3].

Two primary hormones regulate insect metamorphosis: 20-hydroxyecdysone (20E) and juvenile hormone (JH). 20E functions as the principal promoter of molting and metamorphosis by binding to its receptor, activating transcription factors, and regulating downstream gene expression [4]. Riddiford et al. [5] demonstrated that fluctuations in 20E and JH levels determine developmental fate, with JH maintaining larval characteristics by counteracting 20E effects. The equilibrium of this hormonal regulatory network is essential for successful metamorphosis [6].

20E initiates a transcriptional regulatory cascade by binding to its receptor complex, thereby modulating the expression of distinct gene sets during various developmental stages [7]. The 20E signaling pathway can alter chromatin accessibility, creating favorable conditions for transcription factor binding [2]. Simultaneously, JH inhibits the expression of adult-specific genes through its receptor protein Methoprene-tolerant (Met) and downstream transcription factor Krüppel homolog 1 (Kr-h1), while maintaining the larval developmental state by regulating the expression of JH pathway genes [8,9,10,11]. While these studies elucidate the regulatory influence of hormonal signals on chromatin state and gene transcription, genome-wide patterns of chromatin accessibility dynamics during insect metamorphosis warrant further investigation.

Chromatin accessibility analysis techniques have been extensively employed in transcriptional regulation research in recent years. The Assay for Transposase-Accessible Chromatin with high-throughput sequencing (ATAC-seq) has emerged as a predominant method for examining chromatin accessibility due to its procedural simplicity, efficiency, and minimal sample requirements [12]. This technique exploits the preferential activity of Tn5 transposase in open chromatin regions, effectively identifying active regulatory elements throughout the genome [13]. In the context of insect developmental research, ATAC-seq has been utilized to investigate chromatin accessibility fluctuations across developmental stages. Brennan et al. [14] employed ATAC-seq to characterize chromatin accessibility during *Drosophila* embryonic development, revealing dynamic alterations in development-specific regulatory elements. These findings suggest that chromatin accessibility changes are integrally associated with gene expression reprogramming during development.

Despite ATAC-seq’s considerable potential in insect developmental research, comprehensive investigations of chromatin accessibility dynamics during critical metamorphic transitions—particularly the prepupal and pupal periods—remain limited. There exists a significant gap in studies that integrate chromatin accessibility data with transcriptomic profiles (RNA-seq) to elucidate functional relationships between chromatin state alterations and transcriptional regulation.

In this study, we examine chromatin accessibility dynamics during metamorphosis in two representative holometabolous insects, the harlequin ladybird (*H. axyridis*) and the fruit fly (*D. melanogaster*), employing the Assay for Transposase-Accessible Chromatin with high-throughput sequencing (ATAC-seq). We identified differentially accessible regions (DARs) across larval, prepupal, and pupal developmental stages, and characterized their genomic distribution and functional enrichment patterns. Through integration with RNA sequencing (RNA-seq) data, we constructed transcriptional regulatory networks governing metamorphosis in both species. This investigation provides novel insights and comprehensive datasets that enhance our understanding of transcriptional regulatory mechanisms during insect metamorphosis, highlighting the crucial role of chromatin accessibility in developmental processes. Additionally, these findings establish a theoretical foundation for insect developmental biology and may identify potential targets for innovative integrated pest management strategies.

## 2. Materials and Methods

### 2.1. Sample Collection

One mating pair of *H. axyridis* and *D. melanogaster* was selected using CO_2_ anesthesia, and the insects were subsequently reared under the above-mentioned conditions (25 ± 1 °C and 75% relative humidity). Larvae were collected 24 h post-molting. Given that the prepupal phase in both insects lasts approximately 24 h, prepupae were collected 12 h after entering this phase. Pupae were collected 48 h after pupation. Developmental stages were verified using a Sony FDR-AX60 video camera (Sony, Tokyo, Japan). For *H. axyridis*, entry into the prepupal stage was characterized by the cessation of movement and adoption of a curled posture. For *D. melanogaster*, the transition to the prepupal stage was identified when larvae climbed to the upper portion of the tube and their bodies began to contract. The criteria for determining the onset of pupation were consistent with those used for identifying the prepupal stage. At each developmental time point, 10–20 samples were collected for cellular preparation, with two biological replicates per group.

### 2.2. ATAC-Seq Library Preparation

Fresh insect samples were directly snap-frozen in liquid nitrogen and immediately transferred into pre-chilled 1× PBS buffer (pH 7.4). All operations were conducted on ice to maintain cell viability. For nuclei extraction and purification, tissue samples (50–100 mg) were placed in a Dounce homogenizer containing 500 μL of pre-chilled homogenization buffer A (10 mM HEPES pH 7.5, 10 mM KCl, 0.1 mM EDTA, 0.1 mM EGTA, 1 mM DTT, 0.5 mM PMSF, 1× protease inhibitor cocktail). Tissues were gently homogenized on ice with 20–25 strokes using both loose and tight pestles to release intact nuclei while preventing bubble formation. Triton X-100 was added to a final concentration of 0.5%, gently mixed, and incubated on ice for 10 min to promote cell membrane lysis. The homogenate was filtered through a 70 μm nylon cell strainer to remove tissue fragments and aggregates. The filtrate was transferred to a pre-cooled 15 mL centrifuge tube and combined with an equal volume of 2.1 M sucrose buffer (2.1 M sucrose, 10 mM HEPES pH 7.5, 10% glycerol, 0.15 mM spermine, 0.5 mM spermidine, 1 mM DTT). This mixture was carefully layered onto 2 mL of 2.1 M sucrose buffer in an ultracentrifuge tube and centrifuged at 13,000× *g* for 45 min at 4 °C, allowing nuclei to sediment through the sucrose cushion. Following centrifugation, the supernatant was discarded, and the nuclear pellet was gently resuspended in 500 μL of pre-chilled nuclei resuspension buffer (10 mM Tris-HCl pH 7.4, 10 mM NaCl, 3 mM MgCl_2_, 0.1% IGEPAL CA-630, 1× protease inhibitor cocktail) without introducing bubbles.

### 2.3. Nuclei Quality Assessment

For quality assessment of the isolated nuclei, 10 μL of the resuspended nuclei suspension was mixed with 10 μL of 0.4% Trypan blue stain. The mixture was loaded onto a hemocytometer and observed under an optical microscope to examine the morphology and staining pattern of the nuclei.We identified high-quality nuclei based on their round or oval morphology with clear boundaries and confirmed their integrity by ensuring Trypan blue exclusion. The concentration of nuclei was calculated and adjusted to a final concentration of 5 × 10^5^ nuclei/mL.

### 2.4. ATAC-Seq Library Construction

For the initial stage of ATAC-seq library construction, samples were processed using the TruePrep Flexible DNA Library Prep Kit for Illumina (Vazyme, Nanjing, China). Briefly, 5 × 10^4^ nuclei were resuspended in 25 μL of the kit-provided lysis buffer. Following the manufacturer’s protocol, 25 μL of 2× TD buffer and 2.5 μL of the TruePrep Transposase enzyme mix were added to the nuclei suspension. The mixture was incubated at 37 °C for 30 min with gentle agitation to perform tagmentation, integrating adapter sequences into accessible chromatin regions. The resulting DNA fragments were then purified using the kit-provided purification reagents to prepare them for downstream processing.

In the subsequent stage, PCR amplification and index addition were conducted using TruePrep Index Kit V2 for Illumina (Vazyme, Nanjing, China). Purified DNA fragments from the initial preparation were combined with the kit-provided PCR reaction components, and each sample was assigned a unique index sequence to enable multiplexed sequencing. Limited-cycle PCR amplification (10–12 cycles) was performed to amplify the library.

Library quality was assessed by measuring DNA concentration via Qubit fluorometric quantitation and evaluating fragment size distribution using an Agilent 2100 Bioanalyzer (Agilent, Santa Clara, CA, USA). A typical ATAC-seq library exhibits a characteristic periodic distribution pattern corresponding to nucleosome-free regions (<100 bp), mono-nucleosome regions (180–250 bp), di-nucleosome regions (360–500 bp), and higher-order nucleosome regions. When necessary, real-time quantitative PCR was employed to verify library quality by detecting enrichment in specific open chromatin regions.

Validated ATAC-seq libraries were sequenced on an Illumina high-throughput platform (NovaSeq or HiSeq series) using paired-end sequencing (PE150), with a recommended sequencing depth of at least 20 million high-quality read pairs per sample for subsequent bioinformatic analysis. ATAC-seq raw datasets were deposited in the NCBI Sequence Read Archive (SRA) database. The BioProject number is PRJNA982428 (https://www.ncbi.nlm.nih.gov/sra?linkname=bioproject_sra_all&from_uid=982428 (accessed on 11 June 2023)).

### 2.5. ATAC-Seq Analysis

ATAC-seq raw sequencing datasets were initially evaluated using FastQC (v0.12.1) to assess sequence quality distribution, GC content, duplicate sequence proportions, and adapter contamination. Quality filtering was performed with Trimmomatic (v0.39) [15] using the following parameters: adapter sequence removal, trimming of low-quality bases (Q < 3) at both ends (LEADING:3, TRAILING:3), elimination of regions with average quality scores below 15 using a 4 bp sliding window (SLIDINGWINDOW:4:15), and filtering out reads shorter than 36 bp (MINLEN:36). Processed data were subsequently reassessed with FastQC to confirm compliance with quality standards.

Quality-filtered paired-end reads were aligned to the reference genome using Chromap (v0.2.4), which is specifically optimized for chromatin profiling and includes a built-in ATAC mode [16]. This mode integrates multiple preprocessing steps—including adapter trimming, sequence alignment, and PCR duplicate removal—into a single streamlined workflow, eliminating the need for additional tools. The use of Chromap thus improves computational efficiency while ensuring consistent data preprocessing across all samples, allowing up to 4 mismatches. Alignment outputs were converted into standard BAM format using SAMtools (v1.13) [17], followed by sorting and indexing. Alignment quality metrics, including alignment rates and chromosomal distribution, were evaluated using samtools flagstat and samtools idxstats. Mitochondrial alignments and low-quality reads (MAPQ < 30) were excluded from subsequent analyses.

Genome-wide chromatin accessibility patterns were analyzed using deepTools (v3.5.1) [18]. BAM files were converted into normalized bigWig format using bamCoverage with RPGC (Reads Per Genomic Content) normalization. The computeMatrix function was used to calculate read distribution around transcription start sites (TSS, ±2 kb) in reference-point mode and across gene bodies in scale-region mode. Chromatin accessibility heatmaps and distribution profiles were generated using plotHeatmap and plotProfile, respectively, visualizing dynamic changes in chromatin accessibility across developmental stages.

DARs were identified using MACS2 (v2.2.7.1) [19], with larval stage samples as the control group. MACS2 was configured with a significance threshold of q < 0.05—the no-model parameter for ATAC-seq specific analysis—and extsize 200 for fragment length estimation. Stage-specific differential accessibility regions were determined by comparing pupal and adult stages against the larval stage.

Identified DARs were annotated using HOMER’s (v4.11) annotatePeaks.pl tool, categorizing peaks as promoter regions (±2 kb from TSS), introns, exons, 5′UTR, 3′UTR, or intergenic regions. Genes associated with promoter-located DARs were subjected to functional enrichment analysis using Metascape, including GO terms (biological processes, molecular functions, cellular components) and KEGG pathway enrichment, with a false discovery rate (FDR) < 0.05 as the significance threshold. We employed the Benjamini–Hochberg (BH) procedure to control the FDR in our functional enrichment analyses.

Developmental dynamics of chromatin accessibility were characterized using the Mfuzz R package (v2.52.0) [20] for time-series cluster analysis of DARs across developmental stages. These accessibility patterns were integrated with corresponding RNA-seq expression data to examine correlations between chromatin accessibility changes and gene expression dynamics.

### 2.6. Gene Regulatory Network Construction

For gene regulatory network construction, candidates were selected from genes exhibiting correlated patterns of chromatin accessibility and expression as identified through integrated ATAC-seq and RNA-seq analysis. A weighted gene co-expression network analysis (WGCNA, R package v1.70-3) [21] was implemented using previously published gene expression profiles [22]. The network was constructed utilizing a soft-thresholding power determined by the scale-free topology criterion (R^2^ > 0.8) and a minimum module size of 30 genes, based on experimental perturbation. Key transcriptional regulators within the network were identified by cross-referencing selected genes with the InsectTFDB database (www.insecttfdb.com). These transcription factors were designated as core regulatory factors based on their connectivity and module membership values. The final gene regulatory network was visualized using Cytoscape (v3.9.0) [23], with transcription factors represented as core nodes and node size reflecting connectivity degree.

## 3. Results

### 3.1. ATAC-Seq Data Quality Assessment

To elucidate transcriptional regulatory mechanisms during insect metamorphosis, we designed ATAC-seq analyses to profile distinct developmental stages in *H. axyridis*, including second-instar larvae (L2), fourth-instar larvae (L4), prepupae (PP), and pupae (P), as well as comparable developmental stages in *D. melanogaster*. Given that *D. melanogaster* undergoes only three larval instars, we selected wandering larvae (WL) as the terminal larval stage, along with prepupae (PP) and pupae (P). The insert size distribution of ATAC-seq libraries exhibited distinct peaks at ~50 bp and ~200 bp, corresponding to nucleosome-free and mono-nucleosome fragments, respectively, indicating high-quality chromatin accessibility profiling (Appendix A). Sample-specific differences in fragment profiles suggest variability in nucleosome organization or library preparation efficiency.

We assessed the quality and reproducibility of ATAC-seq data through multiple analytical approaches. Spearman correlation analysis of read counts revealed high consistency between biological replicates within each developmental stage. In *H. axyridis*, Spearman correlation coefficients between biological replicates ranged from 0.86 to 0.95 (Figure 1B), while in *D. melanogaster*, coefficients ranged from 0.95 to 1.00 (Figure 1C), demonstrating robust reproducibility across samples. Principal component analysis (PCA) further delineated distinct chromatin accessibility signatures across developmental stages. In *H. axyridis*, the first two principal components (PC1 and PC2) accounted for 88.0% and 9.8% of the total variance, respectively, with samples clustering according to developmental stage (Figure 1D). Similarly, in *D. melanogaster*, PC1 and PC2 explained 84.3% and 11.8% of the variance, respectively, exhibiting clear stage-specific separation (Figure 1E). Scree plots of eigenvalues confirmed that the first three principal components captured the majority of variance in both species, with a pronounced decline in explanatory power after PC3 (Figure 1F,G). The results above demonstrate that our data quality is robust, with strong reproducibility between biological replicates and clear separation in principal component analysis. This high reproducibility provides a solid foundation for the subsequent analysis of chromatin accessibility dynamics.

### 3.2. Chromatin Accessibility Pattern Analysis

To investigate chromatin accessibility dynamics during development, we analyzed ATAC-seq data from critical metamorphic transitions in both *H. axyridis* and *D. melanogaster*. Genome-wide chromatin accessibility profiles in *H. axyridis* revealed distinct stage-specific patterns throughout development. Second-instar larvae (L2) exhibited moderate accessibility levels that increased progressively in fourth-instar larvae (L4), with peak accessibility observed in the prepupal stage (PP), particularly near transcription start sites (TSS). In our previous study, we established the developmental stage correspondence across different insect species and determined that the larval stages of *D. melanogaster* and *H. axyridis* are comparable [22]. Based on this correspondence, we selected the L2 stage as a reference point for differential chromatin accessibility comparisons across subsequent developmental stages.

Chromatin accessibility in *D. melanogaster* displayed developmental stage-specific patterns, with a dramatic increase during the wandering larval stage (WL) compared to the second instar (L2), followed by a gradual reduction during the prepupal (PP) and pupal (P) stages (Figure 2B). These temporal patterns suggest that distinct waves of chromatin remodeling coincide with critical developmental transitions during insect metamorphosis.

### 3.3. Differential Accessibility Regions Analysis

To further characterize chromatin accessibility changes during metamorphosis, we identified DARs by comparing each developmental stage to the reference L2 stage. In *H. axyridis*, DARs predominantly occurred in promoter regions (particularly those ≤1 kb from TSS), with substantial representation in first introns and distal intergenic regions (Figure 3A). Quantitative analysis revealed the highest number of promoter–TSS peaks in the PP vs. L2 comparison (4167), followed by the P vs. L2 (1076) and L4 vs. L2 (334) comparisons, highlighting the prepupal stage as a critical period for chromatin remodeling during *H. axyridis* metamorphosis (Figure 3B). In *D. melanogaster*, DARs were present in markedly higher proportion in promoter regions (≤1 kb) compared to *H. axyridis*, especially in the WL vs. L2 and PP vs. L2 comparisons, with relatively fewer DARs in intronic and distal intergenic regions (Figure 3C). Quantitatively, promoter–TSS peaks were substantially more numerous in early metamorphosis stages, with 3801 peaks in WL vs. L2 and 2239 in PP vs. L2, but dramatically decreased in number to only 105 in P vs. L2 (Figure 3D).

To elucidate the biological relevance of DARs, we performed Gene Ontology (GO) and Kyoto Encyclopedia of Genes and Genomes (KEGG) enrichment analyses for both insect species. In *D. melanogaster*, GO enrichment revealed strong associations between metamorphosis-related processes and the transition from wandering larvae (WL) to prepupal (PP) stages, relative to second-instar larvae (L2). Notably, terms such as positive regulation of cellular processes, cell morphogenesis, and animal organ morphogenesis were highly enriched (Figure 4A). Cellular component categories highlighted cytoplasmic and membrane-associated structures, including cytoplasmic vesicles, the ATPase complex, and ribosomal subunits. For molecular function, terms such as transcription factor binding, enzyme binding, and hydrolase activity predominated.

A similar pattern emerged in *H. axyridis*, where GO analysis revealed significant enrichment in morphogenetic processes, including cell morphogenesis, tissue morphogenesis, and animal organ development, particularly during the L4 and PP stages compared with L2 (Figure 4B). Enriched cellular components at these stages encompassed the Golgi apparatus, cell cortex, and cytoplasmic regions, while enriched molecular functions included DNA-binding transcription activator activity, heterocyclic compound binding, and protein kinase activity. KEGG pathway enrichment analysis corroborated the GO findings. In *D. melanogaster*, the transition from WL to PP featured significant activation of metamorphosis-related pathways, including autophagy, cell cycle, and MAPK signaling (Figure 4C). In *H. axyridis*, metamorphosis-associated pathways were predominantly enriched during the PP stage, particularly insect hormone signaling, Hippo signaling, and Wnt signaling pathways (Figure 4D). Overall, the transition from the late larval to prepupal stages appears to be a critical period for the activation of metamorphic regulatory pathways.

### 3.4. Identification of Conserved Metamorphosis-Associated Genes via Integration of Chromatin Accessibility and Expression Dynamics

Based on the identification of metamorphosis-associated DARs, the prepupal stage in *H. axyridis* and the wandering larval to prepupal transition in *D. melanogaster* were identified as periods of maximal chromatin accessibility changes during metamorphosis. To identify candidate regulatory genes that were active during these critical developmental windows, we performed time-series clustering of chromatin accessibility dynamics using Mfuzz. For *H. axyridis*, we selected genes exhibiting peak chromatin accessibility specifically at the PP stage, while for *D. melanogaster*, we included genes with elevated accessibility across both WL and PP stages (Figure 5A).

In our previous work [22], we characterized the dynamic gene expression profiles of both species, identifying genes highly expressed exclusively at the PP stage in *H. axyridis* and those upregulated during the WL and PP stages in *D. melanogaster* (Figure 5B).

By integrating chromatin accessibility and gene expression data, we obtained a refined set of candidate metamorphosis-associated genes displaying both stage-specific chromatin opening and transcriptional activation. Intersection analysis across datasets revealed 608 genes shared between the two species that met both criteria, likely representing a conserved gene set involved in metamorphosis regulation (Figure 5C). The information for the 608 genes can be obtained from Dataset S1.

Among these conserved targets, we identified four transcription factors: dsx (doublesex), E93 (Ecdysone-induced protein 93F), REPTOR (Repressed by TOR), and Sox14 (Sex box protein 14) (Table 1). These factors, belonging to diverse DNA-binding domain families—DM, Ets, bZIP, and HMG, respectively—represent potential key regulators of metamorphic gene expression programs.

### 3.5. The Metamorphosis-Associated Transcriptional Network Architectures in D. melanogaster and H. axyridis

To further explore the regulatory relationships among the 608 conserved candidate genes, we constructed co-expression networks using weighted gene co-expression network analysis (WGCNA) based on RNA-seq expression profiles. The four identified transcription factors served as focal nodes for module detection, given their stage-specific accessibility and expression patterns in both species.

WGCNA revealed several expression modules in *D. melanogaster*, with four transcription factors occupying central positions within the network (Figure 6). Based on the network topology, *Sox14* and *jim* exhibited the highest intramodular connectivity, functioning as major hub genes in the large co-expression cluster located in the lower section of the network. *REPTOR* also formed an important connection center in this region, while *E93* established an independent module center in the upper region of the network.

The *Sox14*-associated co-expression module, represented by the largest red node in the lower–central part of the network, forms dense connections with numerous genes. Among those visibly connected to *Sox14* are *vri* and *C618H11*. These genes are involved in developmental regulation and biological rhythms. The high connectivity of the *Sox14* module underscores its central role in the transcriptional regulatory network.

The *jim* module, depicted as the second-largest red node, co-constitutes the core of the lower network region alongside *Sox14*. Jim exhibits tight connections with multiple genes, including the visible *C622H8* and several surrounding expression genes. This densely interconnected co-expression pattern suggests that jim likely works in concert with Sox14 during developmental processes.

*REPTOR* appears as a medium-sized red node positioned above Sox14 in the network diagram. Genes associated with *REPTOR* include the identifiable *RhoGEF2* and *Stat92E*, which participate in signal transduction and cellular metabolic regulation. This aligns with REPTOR’s function as a regulatory factor in nutritional and metabolic sensing pathways. *E93* forms a distinct module in the upper section of the network, represented by an orange–red node. Genes connected to *E93* include *Eb* and *TfAP-2*, which are involved in developmental regulation and hormonal responses. The relative independence of the E93 module suggests it may regulate a specific set of genes associated with the ecdysone signaling pathway.

Similarly, in *H. axyridis*, *E93* and *dsx* occupied central positions within the co-expression network, forming high-connectivity modules (Figure 7). The *E93*-centered module was associated with genes such as *fhl1*, *hng2*, *pnt*, *spt5*, and *ur*, involved in transcriptional regulation, developmental patterning, and cellular differentiation, reflecting its conserved role as a key effector in ecdysone-mediated metamorphosis. The *dsx*-associated module showed co-expression with genes regulating morphogenesis and signal transduction, including *fra*, *Mvb12*, *Usp47*, *Lis-1*, and *RalGPS*, suggesting its role in sex-specific development and cellular architecture formation. Additionally, interactions with *CG12104* and *tna* (shown in orange/pink) highlight potential novel regulatory relationships. The findings uncover a conserved core of metamorphosis-related regulators integrated within species-specific co-expression network structures.

In summary, the co-expression network analysis revealed both conservation and divergence in transcriptional regulatory mechanisms between *D. melanogaster* and *H. axyridis*. While both species maintain key hub genes governing metamorphosis (particularly *E93*), the network architecture exhibits species-specific organization, with *D. melanogaster* featuring a more distributed regulatory system across four transcription factors (*Sox14*, *jim*, *REPTOR*, and *E93*), whereas *H. axyridis* demonstrates more concentrated organization around E93 and dsx.

## 4. Discussion

### 4.1. Chromatin Accessibility Dynamics as a Driver of Metamorphosis

This study systematically analyzed chromatin accessibility dynamics during key developmental stages in two holometabolous insects using ATAC-seq technology, revealing the central role of chromatin reprogramming in insect metamorphosis. Our results indicate that the prepupal stage represents the period with the most dramatic changes in chromatin accessibility, a finding highly consistent with previous research on insect metamorphic hormone regulation [3,6].

In *H. axyridis*, we observed that chromatin accessibility peaked during the prepupal stage and subsequently decreased significantly during the pupal stage. This temporal dynamic pattern is closely related to the regulatory rhythm of ecdysone 20E and juvenile hormone JH. According to the classical hormone regulation model, the prepupal stage is a critical period when JH concentration drops sharply while 20E concentration rises [4,5]. Our chromatin accessibility data provides the first comprehensive epigenetic landscape across key metamorphic transitions in *H. axyridis*, revealing stage-specific regulatory dynamics that complement the existing hormone-based model of insect metamorphosis.

Notably, the chromatin accessibility profile in *D. melanogaster* at the L2 stage exhibited a unique distribution, with enrichment peaking near transcription end sites (TESs) rather than the expected transcription start sites (TSSs). While such a pattern deviates from canonical ATAC-seq signatures observed in actively transcribed genes, recent studies have reported that developmentally regulated or transcriptionally repressed genes may display non-canonical accessibility patterns, particularly during early developmental transitions or poised chromatin states [24,25].

One plausible explanation is that during the L2 stage, certain gene sets may remain transcriptionally silent but epigenetically pre-configured, with chromatin structure adopting a “primed” configuration that becomes active in later stages. This is consistent with the hypothesis that chromatin remodeling may precede transcriptional activation during metamorphic transitions [24]. Additionally, TES-proximal accessibility has been linked to paused or repressed transcriptional states in embryonic and early larval tissues [25]. Therefore, we speculate that the TES-biased accessibility pattern in L2 larvae may reflect a biologically meaningful, transitional epigenetic state rather than a technical artifact.

Although the precise regulatory mechanism underlying this pattern remains to be elucidated, the overall consistency of our biological replicates, the high-quality library profiles (Appendix A), and the coordinated stage-specific dynamics observed in downstream stages support the reliability of our ATAC-seq data. Further time-resolved studies focused on earlier larval transitions may help clarify the functional significance of this accessibility signature.

### 4.2. Species-Specific and Conserved Patterns in Metamorphic Regulation

Through comparative analysis of chromatin accessibility patterns in *H. axyridis* and *D. melanogaster*, we identified important evidence for both conserved mechanisms and species-specific adaptations in metamorphic regulation. The 608 shared candidate genes represent a core conserved regulatory network in insect metamorphosis, supporting the view of metamorphosis as a key innovation in the evolutionary success of insects [3].

Four core transcription factors (dsx, E93, REPTOR, Sox14) exhibited similar patterns of chromatin accessibility and expression in both species, strongly suggesting their fundamental role in metamorphic regulation. E93, as a key downstream effector of the ecdysone signaling pathway, has been verified for its conservation across multiple insect species [26,27]. Our study extends these findings by demonstrating that conservation exists not only at the gene sequence level but also at the epigenetic regulatory level, suggesting evolutionary constraints on chromatin dynamics for core metamorphic regulators.

However, the two species showed marked differences in the genomic distribution of DARs. *D. melanogaster*’s DARs were primarily concentrated in promoter regions, while *H. axyridis* showed more accessibility changes in intronic and distal regulatory regions. This direct comparison of regulatory element distribution reveals divergent epigenetic strategies that may underlie species-specific developmental traits despite shared hormonal control mechanisms. These differences likely reflect evolutionary divergence in transcriptional regulatory strategies, and may also be related to their genomic structure and regulatory element distribution characteristics.

### 4.3. Key Transcription Factors in Metamorphic Regulation

Ecdysone signaling is the most well-characterized hormonal pathway governing insect metamorphosis. Upon binding to its receptor complex EcR–USP, ecdysone triggers a hierarchical transcriptional cascade that regulates early- and late-response genes critical for tissue remodeling, cell death, and organogenesis. In this context, **E93,** a known direct target of the ecdysone receptor complex, plays an irreplaceable role in initiating metamorphic transitions by promoting larval tissue breakdown and adult structure formation [26,27]. Our network analysis reaffirmed E93’s central regulatory role, as it occupied a hub position in co-expression modules enriched for genes involved in programmed cell death, hormone signaling, and transcriptional regulation. This supports a model where E93 functions as a master integrator of ecdysone-dependent metamorphic signals across species.

In addition to E93, our study highlighted Sox14 and vri as key transcriptional regulators within metamorphosis-associated modules. Both genes are well-documented direct targets of ecdysone signaling, known to mediate chromatin remodeling and temporal gene expression during the larva-to-pupa transition. Sox14, in particular, is required for cytoskeletal reorganization and apoptotic gene expression downstream of ecdysone pulses [28]. Our data revealed that Sox14’s regulatory module was enriched for genes associated with actin dynamics and programmed cell death, supporting its role as a mediator of tissue reorganization during metamorphosis. Similarly, vri (vrille), a transcriptional repressor, may contribute to the precise timing of morphogenetic events, although its specific functions in the adult transition remain to be elucidated.

As a key regulator of the TOR signaling pathway, REPTOR’s important role in metamorphosis is primarily reflected in nutrient sensing and metabolic regulation [29]. Our chromatin accessibility dynamics data identified REPTOR and Sox14 as central nodes in metamorphic regulatory networks, highlighting the importance of nutrient sensing and cytoskeletal remodeling pathways as critical components of the metamorphic program. We found that the REPTOR module was enriched for genes involved in vesicular transport, energy metabolism, and TOR signaling, highly consistent with the biological processes of large-scale cellular remodeling and energy redistribution during metamorphosis.

Traditionally, *dsx* has been primarily recognized as a key regulator in sex determination and differentiation [30]. In our analysis, however, dsx exhibited coordinated chromatin accessibility dynamics during metamorphic transitions in both *D. melanogaster* and *H. axyridis*, and occupied a central position in the co-expression network. These observations raise the possibility that dsx may have broader regulatory influences during metamorphosis beyond its classical role in sexual differentiation.

Notably, genes co-expressed within the dsx-associated module were enriched for functions related to cell–cell adhesion, morphogenesis, and developmental patterning. While these findings suggest a potential role for dsx in general morphogenetic processes, we emphasize that this inference is based solely on integrative analysis of chromatin accessibility and gene co-expression. Experimental validation, such as genetic perturbation assays, will be essential to confirm this predicted function.

This hypothesis aligns with emerging evidence that sex determination genes can exert pleiotropic roles in development [31]. Nonetheless, we regard our interpretation as hypothesis-generating and have revised the manuscript to reflect the predictive nature of this result.

The WGCNA employed in this study provides a robust framework for predicting regulatory modules and identifying hub genes, such as transcription factors, potentially involved in developmental regulation. However, we acknowledge that these predictions are preliminary, and the functional roles of hub genes remain speculative without experimental validation. The absence of module preservation analyses or perturbation experiments, such as RNAi or CRISPR-based approaches, represents a limitation of the current study. To address this, future research should prioritize functional validation of these co-expression networks through targeted molecular assays and in vivo experiments to confirm their biological significance. These efforts will be critical to elucidating the regulatory mechanisms underlying the observed gene expression patterns and advancing their potential applications.

### 4.4. Integrated Analysis of Chromatin Accessibility and Gene Expression

By integrating ATAC-seq and RNA-seq data, we constructed a more complete regulatory map of metamorphosis. The multi-omics integration approach employed in our study allowed for a more comprehensive understanding of the regulatory dynamics underlying metamorphosis than either approach alone could provide. Time-series clustering analysis effectively identified gene sets exhibiting coordinated chromatin opening and transcriptional activation at specific developmental stages. This multi-omics integration approach not only improved the accuracy of candidate gene identification but also revealed the intrinsic connection between chromatin state changes and gene expression regulation [24].

We observed that changes in chromatin accessibility often precede or coincide with changes in gene expression, supporting the model of chromatin reprogramming as a prerequisite for transcriptional regulation. This temporal relationship between chromatin accessibility and gene expression during metamorphosis provides new insights into the sequence of molecular events driving developmental transitions. However, some genes exhibited patterns where chromatin accessibility was not completely consistent with expression levels, possibly reflecting the influence of post-transcriptional regulatory mechanisms or other epigenetic modifications [27].

Particularly noteworthy is that the 608 conserved genes we identified showed highly consistent patterns of chromatin accessibility and expression in both species, providing an important molecular basis for understanding the conserved regulatory mechanisms of metamorphosis. This core set of genes with conserved chromatin dynamics across species not only advances our fundamental understanding of metamorphosis but also potentially identifies targets for innovative pest management strategies. These genes likely represent the most fundamental and critical regulatory network nodes in the insect metamorphosis process.

## 5. Conclusions

Our comprehensive analysis of chromatin accessibility dynamics during insect metamorphosis in *H. axyridis* and *D. melanogaster* reveals that the prepupal stage constitutes a critical period of chromatin reprogramming, with distinct species-specific patterns in the genomic distribution of differentially accessible regions. Through the integration of ATAC-seq and RNA-seq data, we identified 608 conserved genes exhibiting coordinated accessibility and expression changes, including four key transcription factors (dsx, E93, REPTOR, and Sox14) that form core regulatory hubs within metamorphosis-associated gene networks. E93 emerged as a central regulator in both species, forming modules with genes involved in transcriptional regulation, programmed cell death, and hormone-responsive signaling, while dsx displayed broader functions in morphogenesis regulation during metamorphosis. *REPTOR* and *Sox14* modules were enriched for genes involved in nutrient sensing, energy metabolism, and cytoskeletal reorganization, highlighting the importance of metabolic regulation and tissue remodeling during metamorphosis. These findings advance our understanding of the molecular mechanisms underlying insect metamorphosis and identify conserved regulatory elements with potential relevance to pest control. However, the functional utility of these candidate targets in applied settings remains to be experimentally validated.

## Figures and Tables

**Figure 1 biology-14-00912-f001:**
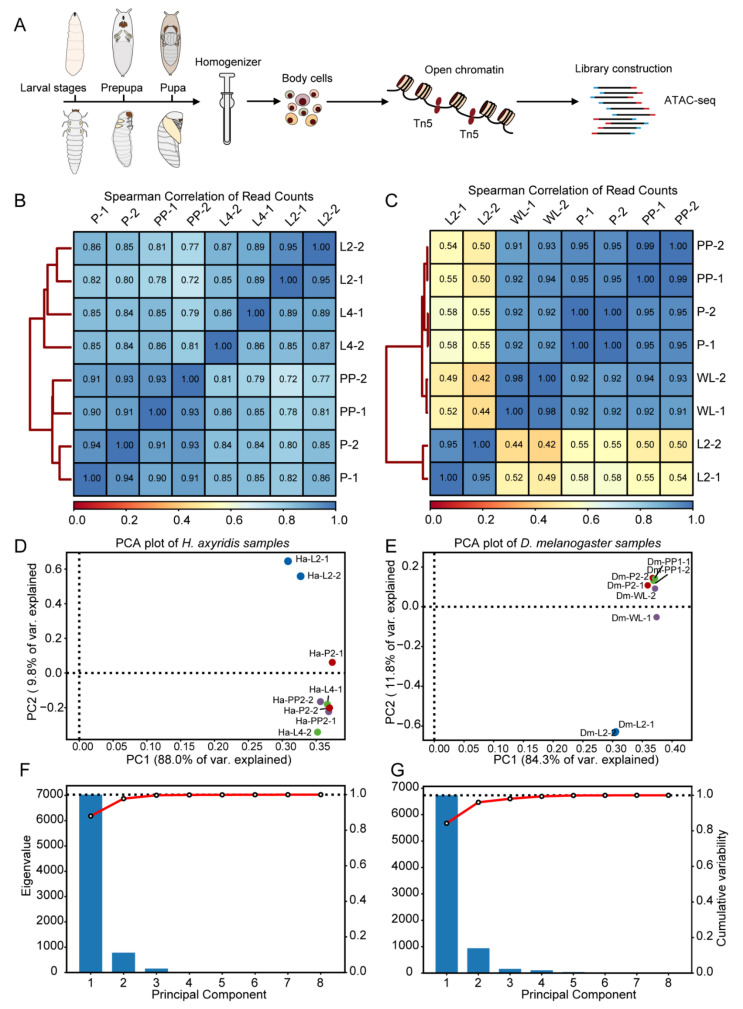
Experimental design and quality assessment of ATAC-seq data. (**A**) Schematic of the ATAC-seq experimental design, showing sample collection and sequencing workflow across developmental stages. (**B**) Heatmap of Spearman correlation coefficients of read counts between biological replicates in *H. axyridis*. (**C**) Heatmap of Spearman correlation coefficients of read counts between biological replicates in *D. melanogaster*. (**D**) PCA plot of *H. axyridis* samples, with PC1 and PC2 explaining 88.0% and 9.8% of the variance, respectively. (**E**) PCA plot of *D. melanogaster* samples, with PC1 and PC2 explaining 84.3% and 11.8% of the variance, respectively. (**F**) Scree plot of eigenvalues for *H. axyridis* principal components, showing the proportion of variance explained. (**G**) Scree plot of eigenvalues for *D. melanogaster* principal components, showing the proportion of variance explained.

**Figure 2 biology-14-00912-f002:**
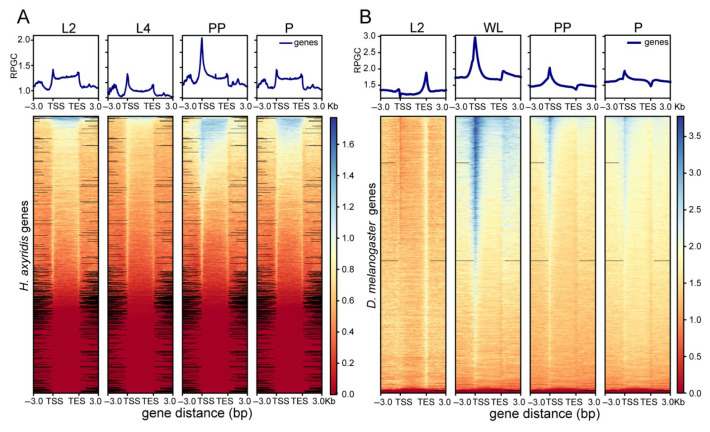
Chromatin accessibility profiles across gene regions in *H. axyridis* and *D. melanogaster* during developmental stages. Chromatin accessibility patterns across gene bodies during development in (**A**) *H. axyridis* and (**B**) *D. melanogaster*. For both species, upper panels show average accessibility profiles at different developmental stages (L2, L4/WL, PP, P) spanning from 3 kb upstream of transcription start sites (TSS) to 3 kb downstream of transcription end sites (TES). Lower panels display heatmaps of accessibility signals for all genes, with color intensity indicating accessibility levels from low (red/orange) to high (blue). For signal normalization, RPGC (reads per genomic content) was employed.

**Figure 3 biology-14-00912-f003:**
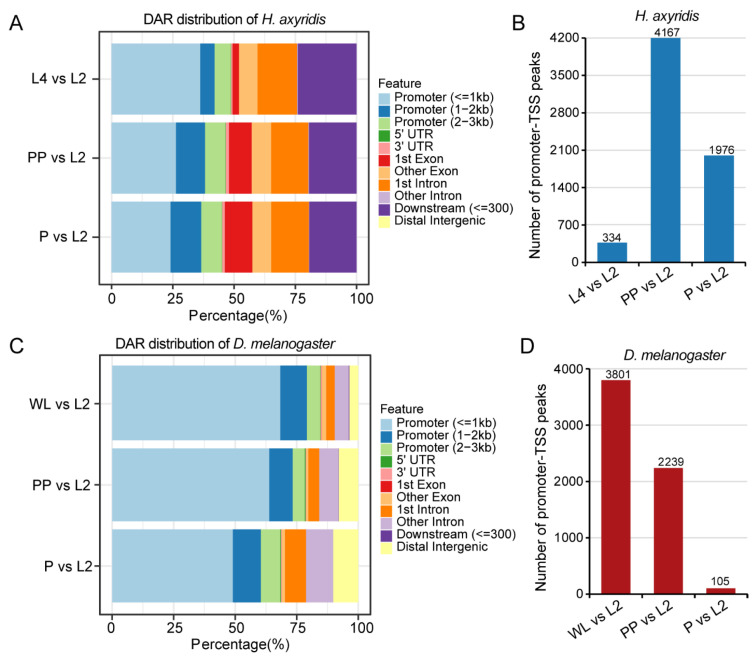
DAR distribution across genomic features in *H. axyridis* and *D. melanogaster*. (**A**) The distribution of DARs across different genomic features in *H. axyridis,* comparing L4 vs. L2, PP vs. L2, and P vs. L2 developmental stages. Each bar represents the percentage of DARs found in specific genomic regions, including promoters of different sizes (≤1 kb, 1–2 kb, 2–3 kb), UTRs (5′ and 3′), exons (1st and other), introns (1st and other), downstream regions (≤300 bp), and distal intergenic regions. (**B**) The total number of promoter–TSS peaks identified in *H. axyridis* across the three developmental comparisons. The highest number of peaks (4167) was observed in PP vs. L2, followed by P vs. L2 (1976) and L4 vs. L2 (334). (**C**) A similar DAR distribution analysis for *D. melanogaster* comparing WL vs. L2, PP vs. L2, and P vs. L2 stages, showing the percentage distribution across the same genomic feature categories as in panel A. (**D**) The total number of promoter–TSS peaks in *D. melanogaster* across the three developmental comparisons. WL vs. L2 showed the highest number of peaks (3801), followed by PP vs. L2 (2239) and P vs. L2 (105).

**Figure 4 biology-14-00912-f004:**
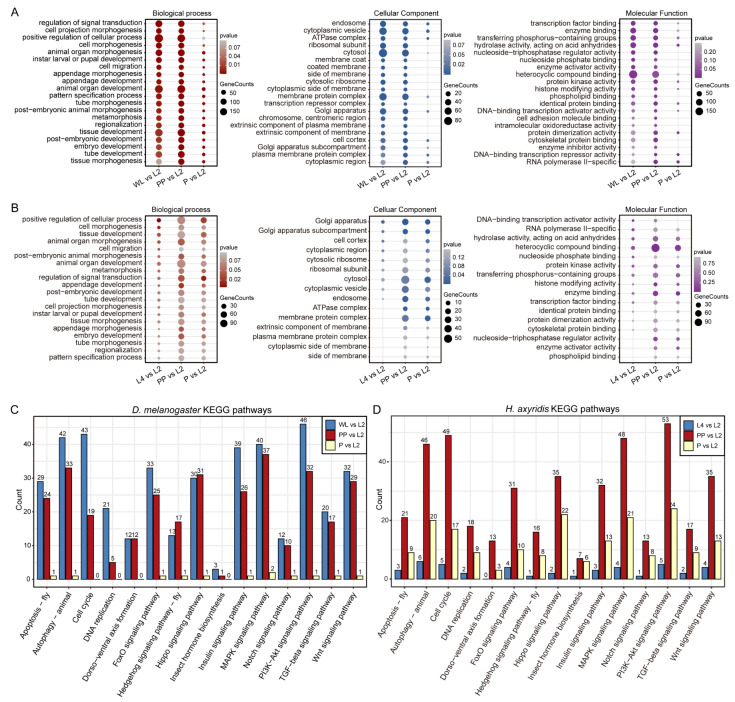
Gene Ontology (GO) and KEGG Pathway Enrichment Analysis in *D. melanogaster* and *H. axyridis*. (**A**) GO enrichment analysis for *D. melanogaster*, showing enriched terms in the biological process, cellular component, and molecular function categories for comparisons between wandering larvae (WL), prepupa (PP), and pupa (P). Dot size indicates gene count, and color intensity reflects *p*-value significance. (**B**) GO enrichment analysis for *H. axyridis*, displaying enriched terms in the biological process, cellular component, and molecular function categories for comparisons between second-instar larvae (L2), (L4), prepupae (PP), and pupae (P). Dot size indicates gene count, and color intensity reflects *p*-value significance. (**C**) KEGG pathway enrichment in *D. melanogaster*, with bar heights representing gene counts for WL vs. P, PP vs. P, and P vs. L2 comparisons. (**D**) KEGG pathway enrichment in *H. axyridis*, with bar heights representing gene counts for L2 vs. P, L4 vs. P, PP vs. P, and P vs. L2 comparisons. Colors distinguish between comparison groups.

**Figure 5 biology-14-00912-f005:**
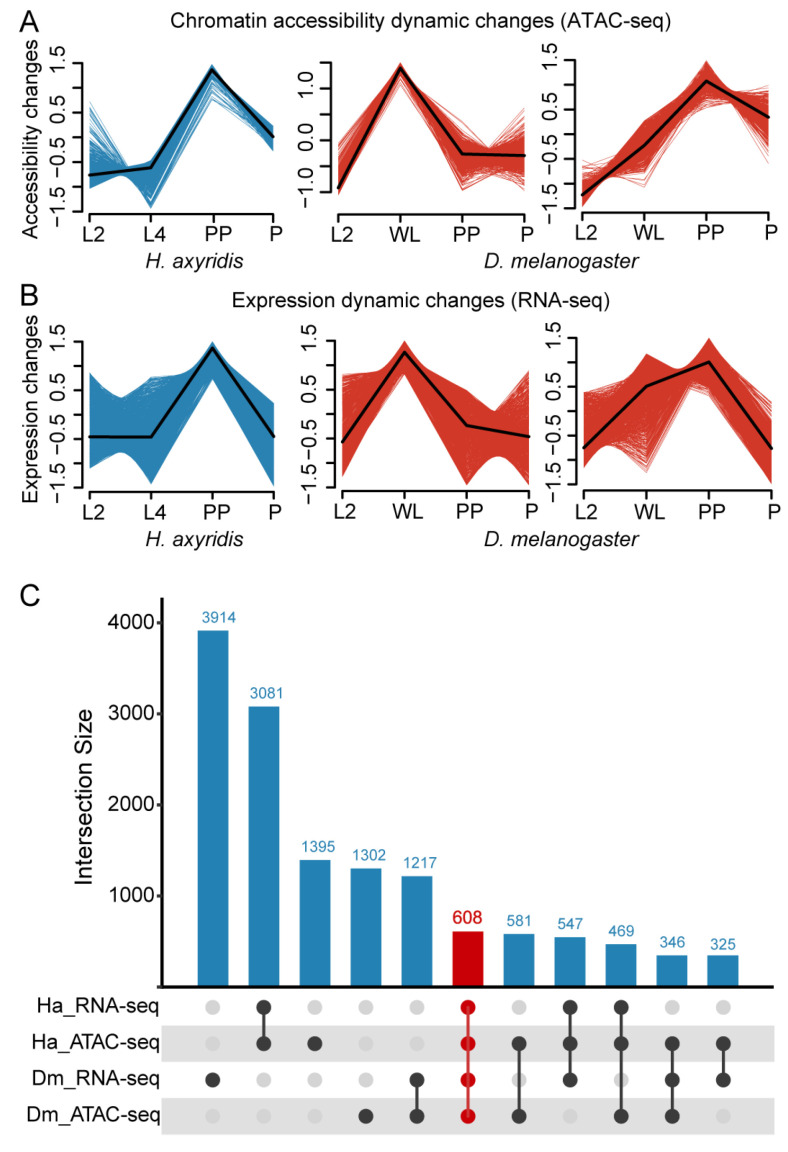
Dynamic chromatin accessibility and gene expression changes during metamorphosis revealed by Mfuzz clustering analysis. (**A**) Chromatin accessibility dynamic changes (ATAC-seq) during metamorphosis in *H. axyridis* and *D. melanogaster*. Accessibility changes are shown across developmental stages: second-instar larvae (L2), fourth-instar larvae (L4), prepupae (PP), and pupae (P) for *H. axyridis*; L2, wandering larvae (WL), PP, and P for *D. melanogaster*. Different-colored lines represent distinct clusters showing similar temporal patterns of chromatin accessibility changes. Shaded areas indicate confidence intervals. (**B**) Expression dynamic changes (RNA-seq) during the same developmental transitions in both species. Gene expression patterns are clustered and visualized with the same temporal framework as those used for panel A, showing coordinated expression changes throughout metamorphosis. (**C**) Intersection analysis of chromatin accessibility and gene expression datasets. The UpSet plot shows the overlap between different combinations of ATAC-seq and RNA-seq datasets from both species (Ha: *H. axyridis*, Dm: *D. melanogaster*). Numbers above bars indicate the size of each intersection, with the largest overlap (3914) observed between *H. axyridis* RNA-seq and ATAC-seq datasets. The connected dots below indicate which datasets contribute to each intersection.

**Figure 6 biology-14-00912-f006:**
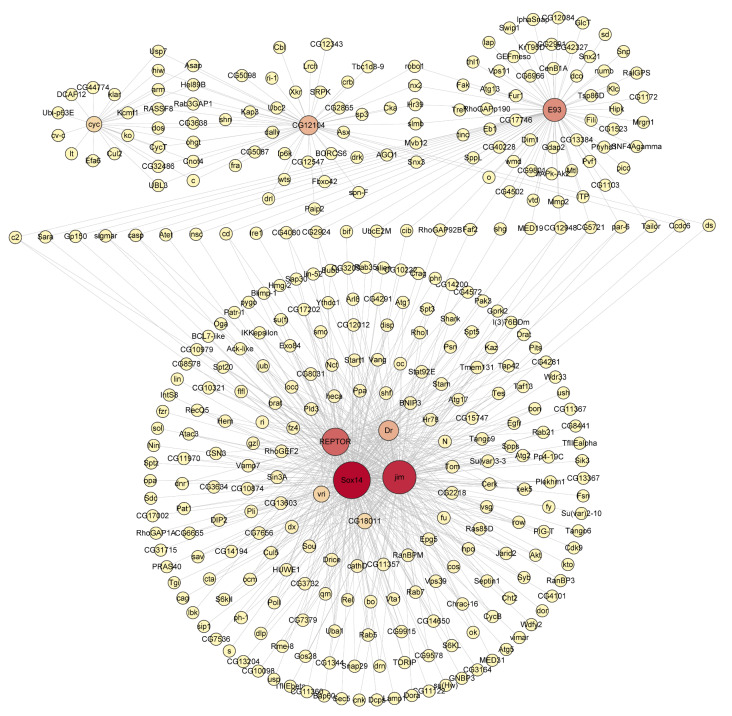
Transcriptional regulatory network of metamorphosis-related genes in *D. melanogaster*. The network diagram illustrates the complex regulatory relationships among key transcription factors and target genes involved in *Drosophila* metamorphosis. The node size indicates the relative importance or connectivity of each gene within the network, with larger nodes representing hub genes with more regulatory connections. The central hub genes highlighted in red (including *E93*, *REPTOR*, *Sox14*, and *jim*) represent key regulatory nodes that coordinate metamorphic gene expression programs.

**Figure 7 biology-14-00912-f007:**
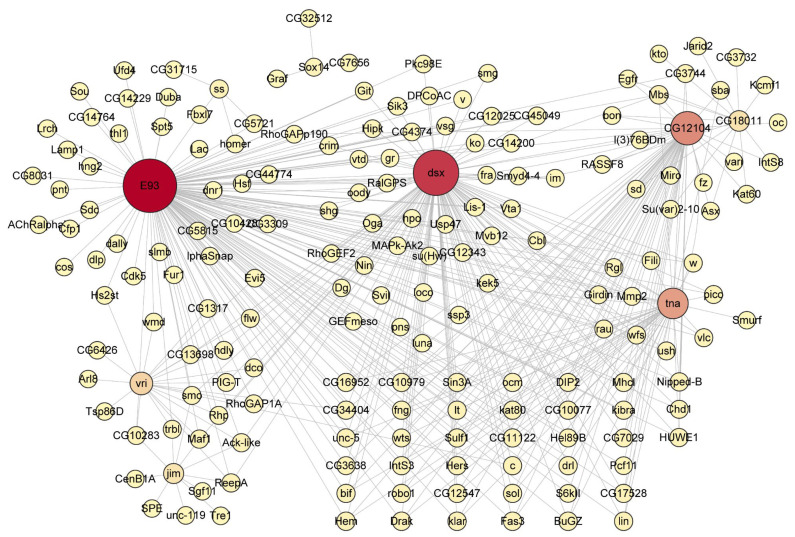
Transcriptional regulatory network of metamorphosis-related genes in *H. axyridis*. The network diagram displays the complex regulatory interactions among key transcription factors and target genes controlling metamorphic development in the Asian ladybird beetle. The node size reflects the relative connectivity and regulatory importance within the network, with larger nodes representing hub genes with extensive regulatory connections. The central hub genes highlighted in red include *E93* (master regulator of metamorphosis), *dsx* (sex determination and differentiation factor), *CG12104*, and *tna*, which serve as key coordination points for metamorphic gene expression programs.

**Table 1 biology-14-00912-t001:** Transcription factors identified within the 608 genes associated with metamorphosis.

Symbol	Name	Domain
dsx	doublesex	DM
E93	Ecdysone-induced protein 93F	Ets
REPTOR	Repressed by TOR	TF_bZIP
Sox14	Sox box protein 14	HMG

## Data Availability

All ATAC-seq datasets are available in the NCBI Sequence Read Archive (SRA) database. The BioProject number is PRJNA982428.

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
