# Peer review of "Chromatin Accessibility Dynamics Reveal Conserved Transcriptional Regulatory Networks During Insect Metamorphosis in Harmonia axyridis and Drosophila melanogaster"

_biology, 2025, doi:10.3390/biology14080912_

Round 1

Reviewer 1 Report

Comments and Suggestions for Authors

This manuscript is aimed at describing the mechanisms that control the process of metamorphosis in two insects: D. melanogaster and H. axyridis. The authors attempted to identify some of the regulatory mechanisms that govern this process by correlating data on changes in chromatin accessibility with changes in the level of gene transcription. These data may be of interest to many readers since not much research is currently aimed at studying the mechanisms of metamorphosis. The advantage of this study is the parallel study of two insects, which may help to identify conservative mechanisms of this process.

The main question for the result of the article related to the ATAC-Seq experiment which was provided for Drosophila L2 developmental stag (Fig. 2 for instance). The problem is that the ATAC-Seq data are significantly differ at this particular stage compared to others (the profile of averaged ATAC Seq signal in L2 does not have peak at TSS of genes as it should, and as other profiles have). Unexpectedly ATAC-Seq signal at L2 is peaks at TES. Normally during the  ATAC-Seq experiments approximately 25% of signal falls into TSS (see for reference https://doi.org/10.1186/s13059-020-1929-3). So, the unusual profile of the ATAS sec signal in L2 larvae makes one suspect the presence of some artifact in this experiment.

It would be very important for authors to clarify this question. It is so important, because all changes in chromatin accessibility in Drosophila genome were assessed by authors relatively this particular L2 stage. So, the change in this only ATAC-Seq, significantly changes the whole message of the manuscript.

Additionally, would be important to clarify in more detail how authors assessed the quality of the obtained ATAC-Seq libraries. Unusually authors used 0.8 volume of AmpureBeads to purify the libraries. This amount can lead to depletion of library fragments of approximately 200 bp in size, which correspond to nucleosome-free regions and typically make up the majority of the library. Do all ATAC-Seq libraries obtained by the authors at different stages have a similar fragment distribution profile? It is interesting that in the methods the authors wrote what a typical library should look like, but they did not write what the libraries they obtained looked like. Was something used in the ATAC-Seq libraries for the signal normalization?

The authors describe in detail how a specific group of genes (608) may be involved in the regulation of metamorphosis. It was strange to see almost no mention in this discussion of the components of ecdysone signaling, which is the best-known regulator of metamorphosis. They discuss the role of the late gene E23, but there is virtually no discussion of the role of ecdysone and the ecdysone receptor. It could be expanded significantly. For instance, two key regulators determined by authors, Sox14 and vri, are the known direct targets of ecdysone.

In addition, authors should be more precise in their use of citations – citing experimental articles when they want to support some important claims rather than reviews. Some quotes that need to be replaced or supported by additional experimental works:

«Yamanaka et al.[4] demonstrated that the 20E signaling pathway can alter…» - 52

«its receptor proteins Methoprene-tolerant (Met) and Krüp» - 55

«Our results indicate that the prepupal stage represents the period with the most dramatic changes in chromatin accessibility, a finding highly consistent with previous research on insect metamorphic hormone regulation[3,6].» - 444

Reviewer 2 Report

Comments and Suggestions for Authors

Tang et al. examine the chromatin accessibility dynamics during metamorphosis in Harmonia axyridis and Drosophila melanogaster by ATAC‑seq and integrate those with published RNA‑seq profiles to discover 608 conserved genes and to reconstruct gene regulation networks based on dsx, E93, REPTOR, and Sox14. Strongpoints of the study include rigorous staging criteria, high Spearman correlations between biological replicates, and comprehensive multi‑omic approach that uncovers conserved and species-specific regulatory details. Weak spots are scarce functional validation of anticipated networks, application of external RNA‑seq metadata with brief method descriptions, and rare mismatch of reported peak numbers. Overall experimental setup and analyses score 85/100 and linguistic clarity 8/10.

The Methods section is thorough in respect to sample handling, nuclei isolation, ATAC‑seq library preparation, and bioinformatic pipelines like FastQC, Trimmomatic trimming, Chromap alignment, MACS2 peak calling (q < 0.05, --no‑model), and HOMER annotation. However, two biological replicates per stage are capable of underpowering differential analysis; duplicate removal by PCR is not well defined; Chromap over other well-used aligners (e.g. Bowtie2) use is not warranted; and RNA-seq integration is short on library prep, read depth, mapping, and normalization. The Discussion rightly relates chromatin motion to hormonal control and stresses the central role of E93 but over-emphasizes the recapitulation of results, disregards possible technical bias (batch effects, variation in genome assembly), and takes unsubstantiated functional claims about dsx and network hubs without experimental follow-up.

The critical issues are:

Utilization of merely two biological replicates for each developmental stage, which may reduce statistical power in DAR identification.

Absence of direct PCR duplicate removal during processing of ATAC‑seq data with regard to artificial peak inflation.

Chromap alignment without cross-validation against benchmarked tools, with potential effects on mapping quality and downstream analysis.

RNA‑seq integration following another study lacking proper methodological clarity (e.g., read depth, normalization strategy), which prevents reproduction

Developmental stage equivalency between H. axyridis and D. melanogaster (e.g., L2 and WL) is assumed but not confirmed by comparative developmental markers.

Gene regulatory network inference via WGCNA is reported without supporting evidence (e.g., module preservation statistics, experimental perturbation), inviting hub functions to overinterpretation.

Functional enrichment analyses report FDR < 0.05 but do not detail correction strategies or multiple testing control beyond defaults.

Claims of novel dsx functions in morphogenesis lack direct experimental evidence and rely solely on co‑expression and accessibility correlations.

Minor other issues are:

Mismatch between count of PP vs. L2 promoter peaks (4167 in Figure 3B vs. 4187 in text).

" WL" (wandering larvae) is not defined at first mention in Section 3.1.

BioProject PRJNA982428 is cited without having a direct SRA URL or accession details for ease of data retrieval.

Table 1 has no criteria for the selection of the four transcription factors and fails to refer to Dataset S1.

Reference numbering is incorrect (e.g., citation [21] in Discussion does not correspond to the reference list).

No representative size distribution plots or gel images for ATAC‑seq library QC.

Impressions of the results: The results evidently describe stage‑specific chromatin remodeling and disclose a conserved central set of metamorphosis‑related genes, with important implications for developmental and evolutionary biology, but functional utility for pest control requiring experimental confirmation.

Reviewer 3 Report

Comments and Suggestions for Authors

To the Authors

The manuscript presents a relevant experimental limitation from the very beginning: the authors chose to anesthetize the insects using COâ‚‚. This method is well-documented to alter gene expression across different species, regardless of the individuals’ age. Previous studies that employed COâ‚‚ report such effects and recommend alternative approaches, such as nitrogen or cold anesthesia. Given this context, I strongly encourage the authors to provide a clear and well-supported justification for the use of COâ‚‚ in their experimental design. This point is particularly critical considering that the method used for chromatin accessibility analysis, ATAC-Seq, is highly sensitive and may further amplify the confounding effects introduced during this early experimental step, thereby affecting the reliability of the results.

Round 2

Reviewer 2 Report

Comments and Suggestions for Authors

After diligent review of the revised manuscript and authors' responses, I am happy to note that the majority of reviewer concerns have been adequately addressed. The authors have improved methodological transparency, clarified interpretations, and corrected technical issues. The significant revisions include justification of biological replicates, confirmation of duplicate removal of PCR, confirmation of Chromap vs. Bowtie2, correct citation of RNA-seq methodologies, and clarification on comparisons among developmental stages. Assertions for the role of dsx and possibilities for pest control have been dampened, and statistical corrections (e.g., FDR) and data availability are now stated clearly. Discussion of WGCNA results now includes the required caveats. There are still two issues that need to be addressed: the speculative nature of pest control applications and the absence of experimental validation of co-expression network predictions. With a few minor copyedits, the manuscript is publishable and has greatly improved overall.

Examples:

Please make sure that Figure 3B correctly represents the comparisons in the text. The present heights and labels do not appear to be in agreement with reported peak counts, particularly for the "PP vs L2" comparison. An analogous test should be performed on Figure 3D.

If both the "TruePrep Flexible DNA Library Prep Kit" and "TruePrep Index Kit V2" are used, describe their functions and maintain naming consistency throughout sections.

Ensure that Figure 4C references the correct stage comparisons. The caption now mentions comparisons relative to "P" when the text implies they are relative to "L2."

Ensure consistency of citation style throughout, e.g., spacing before brackets (e.g., "Riddiford et al. [5]"), period placement, and DOI links (use the https://doi.org. format consistently).

Proofread for minor grammatical issues, such as consistent application of hyphenation (e.g., "co-expressed"), avoidance of unnecessary words, and repetition elimination of organism names in one sentence.

Author Response

Comments 1: Please make sure that Figure 3B correctly represents the comparisons in the text. The present heights and labels do not appear to be in agreement with reported peak counts, particularly for the "PP vs L2" comparison. An analogous test should be performed on Figure 3D.

Response 1: We have carefully reviewed and revised the reported peak counts in the text and cross-checked them against the data presented in Figure 3.

Comments 2: If both the "TruePrep Flexible DNA Library Prep Kit" and "TruePrep Index Kit V2" are used, describe their functions and maintain naming consistency throughout sections.

Responses 2: Thank you for your valuable feedback. To address your suggestion regarding the clarification of the roles of the TruePrep Flexible DNA Library Prep Kit and TruePrep Index Kit V2 in the ATAC-seq library preparation, we have revised the Methods section to explicitly emphasize their distinct functions in separate stages. The revised text now clearly delineates the initial tagmentation and library preparation stage (using the TruePrep Flexible DNA Library Prep Kit) and the subsequent PCR amplification and index addition stage (using the TruePrep Index Kit V2). (Revised in line 146-159)

Comments 3: Ensure that Figure 4C references the correct stage comparisons. The caption now mentions comparisons relative to "P" when the text implies they are relative to "L2."

Responses 3: We have verified that the comparisons in Figure 4C are relative to the second instar (L2) stage, as implied in the text. To address the discrepancy, we have revised the Results section to explicitly state that Figure 4C comparisons are made relative to L2, aligning the text with the figure caption (line 317-323).

Comments 4: Ensure consistency of citation style throughout, e.g., spacing before brackets (e.g., "Riddiford et al. [5]"), period placement, and DOI links (use the https://doi.org. format consistently).

Responses 4: We have carefully reviewed and revised all citations throughout the manuscript to ensure uniformity.

Comments 5: Proofread for minor grammatical issues, such as consistent application of hyphenation (e.g., "co-expressed"), avoidance of unnecessary words, and repetition elimination of organism names in one sentence.

Responses 5: We have carefully checked and revised the manuscript to ensure consistent hyphenation, removed unnecessary words, and eliminated repeated organism names within sentences.
